# Parents’ Communication with Teachers about Food and Nutrition Issues of Primary School Students

**DOI:** 10.3390/children9040510

**Published:** 2022-04-04

**Authors:** Gozde Aydin, Claire Margerison, Anthony Worsley, Alison Booth

**Affiliations:** 1Institute for Physical Activity and Nutrition (IPAN), School of Exercise and Nutrition Sciences, Deakin University, Geelong 3217, Australia; claire.margerison@deakin.edu.au (C.M.); alison.booth@deakin.edu.au (A.B.); 2School of Exercise and Nutrition Sciences, Deakin University, Melbourne 3217, Australia; anthony.worsley@deakin.edu.au

**Keywords:** food and nutrition, communication, parent, teacher, primary school

## Abstract

Parents and teachers have a major influence in the formation of primary school children’s eating behaviours. Although the cooperation of parents and teachers has frequently been recommended in the promotion of healthy eating habits among primary school children, little is known about the communication between these two groups regarding food- and nutrition-related issues. This paper reports findings from semi-structured interviews with primary school parents (*n* = 19) and primary teachers (*n* = 17), as well as findings from a survey of 787 parents in Australia. Audio-recorded interviews were analysed using NVivo and descriptive statistics were calculated for the survey questions. The results indicated that their communications involved various topics, including allergies, lunchbox content, and supervision requests, through diverse communication channels. The risk of offending each other and time scarcity were reported as communication barriers. Parents mainly expected teachers to ensure that their children were given enough time to eat their lunch, teach healthy eating, and be good role models of healthy eating. This study highlights the need to overcome communication barriers between parents and teachers and support teachers in their multifaceted professional roles.

## 1. Introduction

Only 5% of Australian children aged 2–17 years consume the daily recommended amount of fruit and vegetables, and around 40% of their daily energy comes from energy-dense, nutrient-poor food [1]. In addition, one out of every four Australian children aged 5–17 years is overweight or obese [1], and unhealthy eating habits are known to be major contributors to these conditions [2,3]. Moreover, these habits established during childhood are likely to persist into adulthood [4], raising the risk of cardiovascular disease and type 2 diabetes [5]. A poor diet has been linked to a decreased academic performance in children [6], as well as psychological and social problems [7,8].

The formation of children’s eating behaviours involves biological, social, and environmental factors [9]. Social factors include relationships with others, and, in primary school years (age 5 to 12 years olds), parents and teachers are particularly influential [10,11]. The notion that parents and teachers influence children’s eating habits is also consistent with health behaviour theories, such as Bronfenbrenner’s ecological model [12] and Bandura’s Social Cognitive Theory [13], which recognise that significant adults, such as parents and teachers, affect children’s behaviour through role modelling, normative practices, and social support. Whilst parents are not the only role models for their children, they do control their children’s food consumption by shopping, cooking, and providing food in the home environment [14]. Australian parents also provide food for their children to take to school [15]. In addition, teachers can influence primary (elementary) school children’s eating behaviours as they are responsible for the delivery of food and nutrition education (FNE) in class and spend a significant amount of time with these children [16]. Primary school students have been reported to trust their teachers as health ‘experts’ [17].

The cooperation of parents and teachers has frequently been recommended in the promotion of healthy eating habits among primary school children by researchers [18,19]. For example, if parents and teachers work together to create a consistent approach to improving the diet of children, it could ensure that primary school children do not receive mixed messages, which are believed to limit the effectiveness of healthy eating programmes [19,20]. However, little is known about the current interactions and communications between parents and teachers regarding food and nutrition issues in primary schools.

There has been some research on how parents and teachers interact or communicate with children regarding food- and nutrition-related issues [21,22,23,24]. In addition, a few studies in preschools have explored the interactions and communications of parents with teachers about food and nutrition issues [25,26,27,28]. However, parents’ and teachers’ communications and interrelationships regarding food- and nutrition-related issues in primary schools have generally been underexamined. This paper aims to fill this gap. As the effectiveness of any healthy eating initiative among school children relies on collaboration between stakeholders, including parents and teachers [20], the examination of barriers, facilitators, and opportunities for better communication between these stakeholders can support both groups in promoting healthy eating among children.

## 2. Materials and Methods

The study was underpinned by stakeholder theory [29]. In the field of health promotion, stakeholder theory has recently gained prominence [30,31]. The theory can be used to bring about changes at the organisational and policy level to address public health issues [31]. Stakeholder theory helps in the understanding of how the behaviour, intentions, agendas, interrelations, and interests of stakeholders can impact decision making [29]. In the present context, teachers and parents are primary stakeholders.

Our research employed exploratory sequential mixed methods by combining qualitative and quantitative data collection and analysis in two studies [32]. We conducted the qualitative study first to identify key themes and questions requiring further exploration. The results of the qualitative study informed the subsequent quantitative study and helped to develop the instrument (survey). In addition, the quantitative data helped us check the generalizability of the initial qualitative results [32].

### 2.1. Study 1. Qualitative Study—Interviews with Parents and Teachers

Qualitative semi-structured interviews were conducted with parents and teachers. The interview questions were original and based on a review of the literature in which gaps were identified. This paper reports on parents’ and teachers’ responses to a single broad question included in these interviews: ‘How would you describe your relationship and communication with parents/teachers regarding food and nutrition related issues?’

#### 2.1.1. Participants and Recruitment

Parents and teachers were purposefully [33] recruited from Melbourne suburbs of varying socioeconomic levels and regional areas of Victoria. The inclusion criterion for parents was having a child attending any primary school in Victoria, whereas, for teachers, it was having taught any aspect of food and nutrition in any primary school in Victoria. Study advertisements were circulated on parent and teacher Facebook groups, and notices were posted on neighbourhood notice boards at public places. A snowball sampling strategy [34] for recruiting parents was used, where participants helped researchers to identify other potential participants. Thirty-seven parents and twenty-six teachers expressed their interest in participating in the study via email. The name and suburb of their primary school were asked to ensure that no more than one parent/teacher was included from any particular school. This step assisted in ensuring that parents and teachers represented all of government, Catholic, and independent schools, including religious schools, such as Christian, Islamic, and Jewish schools.

A plain language statement and consent form were sent to all the parents and teachers selected for the interview via email before each interview. More parents and teachers from government schools volunteered to take part in the interviews than were required. Therefore, eighteen parents and nine teachers from government schools were not invited to participate and were informed by email. All participants gave their written informed consent to participate in the study and permission to audio-record the interviews. Participants who completed the interviews received a $20 shopping voucher for their time. The researchers had no prior relationships with the interviewees. The recruitment was finalised when the ‘conceptual depth’ (data saturation) point was reached [35]. This is the point when further data gathering and analysis added little that was new to the conceptualisation [36].

#### 2.1.2. Interview Procedure

Semi-structured qualitative individual interviews took place either face-to-face (*n* = 15) or via phone (*n* = 21) between January 2020 to May 2020 by the first author (GA). Face-to-face interviews took place in public places such as libraries or at participants’ homes. Parental interviews lasted between 22 and 54 min, with an average of 37 min. Teachers’ interviews lasted between 28 and 62 min, with an average of 47 min. The reported interview durations included responses to all four questions of the interview; only one of which is reported in this paper. 

Two pre-test interviews for each group of participants were used to assess the face validity of the interview questions. Since only small language changes were made to the questions after pre-testing, the pre-tested data were combined with information obtained from further interviews. To let interviewees explain more, the interviewer used general prompts such as “Could you tell me more about that?”, “Is there anything else?”, “Why?”, “What do you mean by …”, “How…?”/“Where…?/“What for…? 

#### 2.1.3. Analysis of Qualitative Study Data

An external contractor (Rev.com (accessed on 25 January 2020)) transcribed all interview recordings verbatim, and GA checked the transcripts for accuracy. All participants were offered the opportunity to review their interview transcripts, but only one teacher accepted the invitation and subsequently made no changes. Interview transcripts were coded using the qualitative data analysis software NVivo (QSR International, Doncaster, Australia, Version 12). GA familiarised herself with the data by reading the transcripts multiple times. The data analysis began as soon as the first two interviews were transcribed, using King’s template analysis method [37]. Based on the research question, an initial template with a priori codes was constructed. This was utilised to expedite the interview coding process [37]. During the remaining data coding process, new codes were introduced into the initial template. During weekly meetings, the codes detected by GA and the research context were examined with other study researchers for accuracy of interpretation [38]. All authors reviewed the final template that contained the themes.

### 2.2. Study 2. Quantitative Study—Survey of Parents/Primary Caregivers 

#### 2.2.1. Participants and Recruitment

Respondents were eligible to participate if they were currently living in Australia and were the parent or primary caregiver of a child attending an Australian primary school. Deakin University’s social media channels, as well as Facebook and Twitter, were used to promote the study. Respondents could complete the survey using the link provided in the online adverts between March and April 2021. The Qualtrics platform was used to deliver an online cross-sectional survey.

Ethics approval was granted by the Deakin University Health Human Ethics Advisory Group (Project No: HEAG-H 147-2019 and HEAG-H13-2021).

#### 2.2.2. Survey

The qualitative interviews conducted with parents and teachers (as described above) informed the development of the survey. This paper reports findings from four closed-ended questions from the survey regarding parents’ communication with classroom teachers about food- and nutrition-related issues, as well as their expectations of teachers. Questions and response options are shown in Table 1.

#### 2.2.3. Analysis of Quantitative Survey Data

The responses to the survey questions were analysed using IBM SPSS Version 27 (Chicago, IL, USA). Descriptive statistics were calculated, including frequencies and percentages.

## 3. Results

### 3.1. Findings from the Qualitative Study: Parent and Teacher Interviews

A total of 19 parents were interviewed. Despite their different ethnic backgrounds (13 parents were from Chilean, Turkish, Israeli, Japanese, Indonesian, New Zealander, Singaporean, Italian, Vietnamese, Polish, Persian, Swedish, and Taiwanese backgrounds, whereas six had an Australian background), all of the parents interviewed had lived in Australia for at least 6 years. They were all post-secondary educated and stated that they worked at least part-time. They all had at least one child (aged 5–12 years old) enrolled in a primary school in Victoria. Seventeen teachers who were previously employed (*n* = 2) or were currently working in a primary school in Victoria (*n* = 15) were interviewed. The duration of teachers’ experience varied considerably: six had between zero and four years, three between five and nine years, three between ten and fourteen years, two between fifteen and nineteen years, and three had over nineteen years of experience. Teachers were recruited from the suburbs of Melbourne and the regional areas of Victoria. Full details of the demographic characteristics of the interview participants have been presented previously [39].

Although most parents reported limited communication with teachers regarding food- and nutrition-related issues, several themes emerged from the interviews (Table 2). The interviews revealed that the topics parents and teachers discussed included allergies, insufficient food in lunchboxes, and lunch supervision requests. Communication methods and barriers were also discussed, and interviewees provided a number of suggestions to improve the communication regarding food- and nutrition-related issues.

#### 3.1.1. Theme 1. Topics Discussed

This theme encompassed specific topics that parents and teachers were concerned about, but did not include the general communications that schools sent to parents, such as the ‘nude food’ policy [unpackaged food policy] and rules expected to be followed at lunchtime. 

Allergies: In some schools, allergies were managed through a school-wide policy, whereas other schools preferred to manage these situations by class-specific rules. In the latter situations, several parents (*n* = 7) and teachers (*n* = 8) mentioned that parents were informed about allergy issues in their children’s classroom and were expected to provide food other than those particular allergenic foods.

‘If there is a student in one class who has a severe nut allergy, then that information needs to be communicated, and parents need to be mindful of what they’re sending in their children’s lunch boxes’Teacher 7

Parents confirmed that teachers supported them and took care of their children when they wanted their children to avoid certain types of food due to allergic reactions. For example, a parent said:

‘If you tell them that you don’t wish your kid to have a certain type of food or not to share food with other students, they are quite aware of that and they are taking care to go according to the parents’ wishes’Parent 8

Insufficient food: Teachers (*n* = 4) and parents (*n* = 4) also stated that a dialogue occurred when food was consistently insufficient for a child or forgotten to be provided on a particular day. 

‘We did have a boy last year who was malnourished because his nanny was preparing his lunch and wasn’t putting enough food in his lunch break, he was always hungry. So, we told the mother and they doubled the amount of food and she was unaware of it that the nanny was putting in so little food.’ Teacher 5

‘The only time you hear from the school is if the kids don’t have the lunch box.’ Parent 2

Teachers also stated that they let the family know if school had provided some food to the student. 

‘If we provide them with lunch because they haven’t brought any, a note will go home to say, “We provided your child with this today, due to the fact of not having any lunch.”Teacher 11

Supervision requests: Parents’ requests for supervision for their children who had various eating issues, including psychological issues and dieting, was also an emerging theme. Parents (*n* = 7) reported requesting or expecting monitoring and encouragement from the teacher, especially for their younger children. Teachers’ responses also supported these requests from parents. For example, a teacher stated: 

‘Most of my communication with parents are when they’ve got concerns with their child’s eating. With young girls and what they’re consuming or not eating’Teacher 16

A parent with a child who had just started primary school mentioned that she requested some supervision from the teacher for her child: 

‘I expressed my concerns with him not eating and that he needs to be supervised because he’s a grazer and he gets easily distracted. So, she said, “I’ll keep an eye on him at lunchtime, just to make sure.”Parent 6

A parent of a grade four student mentioned that she requested teacher’s assistance with her daughter’s eating problems:

‘I emailed last year saying that my child has got anxiety and she was coming home not eating her lunch. So, I emailed the teacher, and they emailed back and made sure that she was eating her lunch’Parent 18

#### 3.1.2. Theme 2. Communication Methods and Barriers

Communication channels: Several communication methods were cited by parents and teachers, such as newsletters, school apps, emails, brochures, face-to-face meetings, and workshops. Newsletters were the most commonly used method to disseminate food- and nutrition-related information to parents. For example, two parents commented:

‘During the induction, there was a one-hour talk about lunchboxes, what food to prepare. Yes, I thought it was pretty good. They gave us the updated food pyramid’Parent 17

‘They also do send home articles to parents about healthy eating. And within the newsletter, they might have an article’Parent 11

Communication barriers: Parents and teachers tried to avoid offending each other, which affected their communication. Some teachers (*n* = 7) recognised the potential of offending parents as a barrier to their communication regarding food and nutrition issues. Similarly, two parents expressed the same concern when sharing their opinions with teachers about rewarding children with sweets.

‘They could get very defensive, and that could impact your relationship. So, it’s a very fine line to talk about food with parents.’ Teacher 14

‘I thought once to send an article about giving lollies as a prize, but I thought it was tricky. I don’t know how the teacher could react to that’ Parent 3

Another barrier identified was the busy nature of contemporary lifestyles. Some teachers (*n* = 4) felt that parents would be too busy to discuss food-related issues or attend a school meeting on food and nutrition.

‘It (healthy eating workshop) was after school one day. It could just be that it was just a timing thing because I think many families have busy schedules, and they tend to head off to activities right after school.’Teacher 7

Suggested ways to improve communication: Almost all parents (*n* = 17) expressed their willingness to attend a healthy eating workshop at their children’s school.

‘It would be good to not make it a one-off event. Maybe parents would come in once a quarter or whatever to talk about food, and maybe even to mini-classes would be pretty cool.’Parent 17

In addition, one parent reported that she requested such a workshop from her children’s school:

‘I’ve actually suggested them, one of the teachers already… to have one of these programs that parents and students can participate and listen to. But nothing has come of it yet’Parent 16

Teachers, on the other hand, believed different communication tools and language should be utilised to get parents engaged. Most of the teachers suggested the increased use of newsletters to disseminate healthy eating information to time-poor parents.

‘Maybe something in the newsletter, like a little healthy eating corner of the newsletter with a little tip or a recipe. Parents can see but it’s not someone telling them. They’re just reading it. I guess it might plant a seed’Teacher 1

‘I think we need to utilise the tools that we have, but really effectively. It depends on your clientele and your school community. Photographs might work really well for a community that don’t speak English, to see what they’re child is doing, for them to understand, because they have no idea what’s going on in the classroom.’Teacher 9

Tailored approaches for the delivery of food- and nutrition-related information for the parent profile of the school were highlighted by a number of teachers. For example, a teacher from a school in an affluent suburb commented:

‘My sister’s a dietician, and parents have just said, “Can you just ask her what I should be giving my kids?” And I’ve talked about workshops and, parents would love it. This is Toorak [a very affluent neighbourhood in Melbourne, Australia]. Toorak would take it on board. Other schools, if you said, “Healthy food workshop,” I think that they would go, “Hell, no.” I think if you framed it differently and had a workshop on building resilience or if you just sold it in such a way that it was about just calming your kid down so they’re not all over the place and they can study better.’ Teacher 14

Another teacher highlighted the importance of building rapport. She commented:

‘Obviously with my school, because they’re food insecure, you might have to connect with some community groups that provide free, cheap food …, make connections with those first, get them very confident in that space first, in asking, because a lot of them don’t ask for help either. They’re too proud, or they’re incredibly embarrassed. You’ve got to bridge the gap through other means to start that conversation, rather than going, “Right. We all have to learn about a healthy lunchbox. We all have to feed our children like this.” We’ve got to be really careful about how we go about it, because we are adding to the stigma. We’re adding to the discrimination and the poverty.’Teacher 12

### 3.2. Findings from the Quantitative Survey: Parent Surveys

Twelve hundred and fifty-nine respondents clicked on the survey link, and seven hundred and eighty-seven completed the survey (62% completion rate). The parents who responded represented a wide range of age, education, geographical regions, and socio-economic level categories. Almost all (96%) were female, and 86% were married. Their mean age was 40 years. Most parents had at least a university degree (72%). Although the survey had respondents across Australia, more than half were from Victoria (56%). Sixty-six per cent of parents lived in major cities, similar to the distribution in the Australian population, where 71% are from major cities [40]. The main language spoken at home was predominantly English (93%). The majority were from high SES (54%) and mid-SES (37%) backgrounds according to the identification by residential postcodes mapped to the Socio-Economic Indexes for Areas (SEIFA). Full details of the demographic characteristics of the respondents have been previously presented [41,42].

The results of the survey showed that most parents (71%) were not contacted by teachers about food- and nutrition-related issues. Some parents (21%) were contacted when there was an allergy issue in the classroom. Parents who chose the other option for this question listed several reasons, including praise for the prepared lunch, too much food in the lunchbox, and sometimes teachers wanted to know about food prohibitions that may have existed at home. The most common reason for parents (36%) contacting teachers was when their child did not eat his/her lunch and needed supervision. Most parents who opted for the ‘other’ option for the question ‘When would you contact your child’s classroom teacher regarding food and nutrition related issues?’ mentioned food allergies. Other issues raised included lunchtime being too short and the use of candies as rewards. More than three-quarters of all parents expected teachers to teach healthy eating, make sure enough time is given to eat lunch, and be a good role model by modelling healthy eating behaviours. Parents believed the single most important action for teachers was to ensure children were given enough time for lunch, and the least important was to give parents feedback on the lunchbox (Table 3).

## 4. Discussion

### 4.1. What Do Parents and Teachers Discuss? 

Both the interview and survey responses suggest that allergies are an issue many parents and teachers discuss frequently. As Australia has one of the highest rates of food allergies in the world, they are one of the main concerns in Australian primary school settings, mainly due to the severe consequences of anaphylaxic reactions, including the death of children [43]. In a recent study conducted with 27 Australian schools, 25 indicated that they had students enrolled with one or more food allergies [15]. In another recent Australian study, 71% of parents of primary school children with a diagnosed food allergy reported regular communication with the classroom teacher regarding food allergy management at their school [44] The severe consequences of food-allergy-related incidents have resulted in the dominance of food allergy management discussions in the food- and nutrition-related conversations between parents and teachers. Whilst this time could be spent on more fundamental food- and nutrition-related issues, it has been suggested that teachers can use this topic as a bridge to introduce nutrition knowledge in their communication with parents in childcare settings [45], which can also apply to the primary school setting. When designing any future food and nutrition information resources targeting parents, allergy issues should also be considered.

The communications between parents and teachers about lunchboxes were relatively sparse, considering around 85% of children rely on parent-provided packed lunches in Australia [46]. Teachers in our study preferred to only contact parents when the food provided was insufficient. Some parents also reported that teachers might send notes to parents regarding the content of the lunchbox. However, our findings indicated that only a few parents expected the teacher to provide feedback on the lunchbox content. Supporting the findings of this study, we have previously reported that many parents thought this practice would be inappropriate and unhelpful [41]. However, previous research has also demonstrated that Australian families struggle to pack lunchboxes that are nutritious, safe, and quick to assemble [47]. Therefore, whilst teacher–parent communication over lunchboxes can be uncomfortable for both groups, alternative ways to assist parents with lunchbox preparation should be considered to help parents who find this task difficult.

A recent review reported the availability of 12 websites in Australia that support parents in preparing sufficient and healthy lunchboxes [48]. For example, the website created by Cancer Council Australia has an interactive tool that helps parents create a lunch box and also check the age-appropriate portion sizes for their children [49]. Another initiative is Nutrition Australia’s ‘Reclaim your lunchbox’ workshop, which is offered free to schools, where parents can learn how to prepare better lunchboxes [50]. However, the reach of these resources and programs is unknown, and efforts to increase their reach is imperative. Schools and teachers could direct parents to these resources to help them comply with school food policies and prepare healthier lunchboxes.

### 4.2. Communication Barriers and Ways to Improve Communication

The risk of offending each other and lack of time were reported as barriers that hinder communication between parents and teachers. These barriers are not unique to Australia or the primary school setting, as they have also been identified in preschools and primary schools in the US [16,45]. Teachers and parents can and should be supported to improve their communication with each other; however, Saltmarsh et al. [51] reported that the discussion of parent–school engagement was largely missing in Australian teacher pre-service programmes. International scholars suggest that teachers should be supported through pre-service and in-service training to develop their social and communicative skills to cooperate effectively with all types of parents [52,53].

In addition, time barriers due to the busy lifestyles of contemporary parents and teachers’ struggles with heightened pressure to perform standardised testing, intensified workloads, and a broadening of the teacher role with additional responsibilities [54] should be addressed. New communication tools that many schools have started to use, such as school apps, can help to overcome time barriers by facilitating fast information exchange [55]. Moreover, supporting classroom teachers with a dedicated school health and wellness team would decrease the teachers’ workload and increase the communication effectiveness with parents.

Teachers, in particular, highlighted the need for tailored approaches to varying parent profiles and using different communication methods wisely. In line with our findings, it has been shown that teachers and parents of different types of schools had different communication channel preferences, and discrepancies between the preferences of teachers and parents of the same school existed [52]. Similarly, in another American study, parents suggested the use of bilingual print and images to address literacy issues [56]. The Family-School Partnerships Framework’ recommends developing an understanding of parent communication needs and preferences, as well as their interests and goals via focus groups, conversations, and surveys [57]. In line with this framework, our findings indicate the need for a consensus on the most appropriate communication methods in school settings.

### 4.3. Parents’ Expectations of Teachers

Many parents believed the most important action that teachers need to take is to ensure that their child is given enough time to eat their lunch. This echoes the findings of a recent study where Australian parents stated lunchtimes were not long enough [58]. Although it is not a decision that teachers make solely, previous studies have described how Australian children have had to eat their food within tight time constraints [59]. Even though a minimum of 20 min is recommended [60], Australian children are usually allowed 10 min [58]. Therefore, schools may need to review the time allocated to eat lunch, which could reduce parental concerns.

The results of our study indicated that teachers are likely to find themselves in a position to intervene in children’s eating practices due to the supervision requests from parents. They are expected to have an active and multifaceted role in food and nutrition in primary schools; however, Australian teachers’ training for this role is insufficient [61,62]. Therefore, this may have counterproductive consequences if teachers are not educated on proper feeding practices, nutritional needs, and requirements. For example, in a Swiss study, parents reported that their children ended up vomiting when they were forced to eat some food they disliked by their supervisor at school [63]. Teacher training opportunities would help teachers to become good role models of healthy eating and help build confidence and knowledge on ways to communicate healthy eating information [64,65].

### 4.4. Further Implications

Although the study was conducted in Australia, the following recommendations still apply to other countries that are combatting rising childhood obesity rates and nonoptimal nutrition issues among young children. Our findings highlight the need for relevant resources and training opportunities provided by public health and education authorities to facilitate parent–teacher communication regarding food- and nutrition-related issues. Existing guides for schools to build and improve family–school partnerships, such as ‘The Family-School Partnerships Framework’ [57] can be extended in partnership with public health authorities to cover effective ways of communicating regarding food and nutrition. In addition, school management teams should support teachers, and dedicated teams should be constructed to assist teachers in food- and nutrition-related communications with parents. Moreover, schools should have a consistent stance backed up with evidence-based information in food- and nutrition-related policies and practices to reduce the conflicts between parents and teachers.

### 4.5. Strengths and Limitations

The researchers who conducted the present study were based in Victoria; therefore, this might have facilitated the recruitment of a higher number of parents from Victoria due to familiarity, personal networks, and access to the Deakin University’s social media profile (which is located in Victoria). However, the main strength of this study was its focus on the views of teachers and parents from various demographic backgrounds across Australia concerning their communications about food- and nutrition-related issues The findings are novel, as this is the first study to explore the food and nutrition aspects of their communications. However, several limitations in the current study must be considered when interpreting the results. Firstly, a self-selecting sample of teachers and parents participated in the studies, reducing the generalizability of the results. Teachers and parents with a greater interest in the topics are likely to have become involved in the study. Secondly, because of limited time and resources, only parents were recruited for the survey part of the research. Furthermore, the advertisements to participate in the survey were only posted on social media. This would limit our pool of potential respondents to only those who follow these platforms. Due to the cross-sectional nature of the study, causal relationships between the variables could not be examined. Another limitation is that the sample was predominately female. Lastly, the views of only two of several stakeholder groups were investigated; the views of other groups, such as school principals, canteen staff, education department administrators, and food companies, need to be investigated in future work.

## Figures and Tables

**Table 1 children-09-00510-t001:** Survey questions and response options.

Have you ever been contacted by your child’s classroom teacher about the following (Tick all that apply)
Not enough food in my child’s lunch box; Certain foods not allowed in my child’s lunch box; When there is a food allergy issue in the classroom; When my child does not eat his/her lunch; I have not been contacted; Other food-related issues, please specify.
When would you contact your child’s classroom teacher regarding food and nutrition issues? (Tick all that apply)
When my child does not eat his/her lunch and needs supervision; When I do not want my child to be exposed to a particular food; To ask what to put in the lunch box or whether a certain food is appropriate to be sent to school; I would never contact my child’s classroom teacher; Other.
To what extent do you agree or disagree with the following statements?
I expect my child’s teacher:
To make sure my child eats all his/her lunch; To make sure my child is given enough time to eat lunch; To give feedback to me about the content of the lunchbox; To make sure school policies on healthy eating are always applied; To be a good role model by modelling healthy eating behaviours; To teach healthy eating.
Response options: Strongly disagree, Disagree, Neutral, Agree, Strongly agree, Unsure
Which single action is the most important for you?
To make sure my child eats all his/her lunch; To make sure my child is given enough time to eat lunch; To give feedback to me about the content of the lunchbox; To make sure school policies on healthy eating are always applied; To be a good role model by modelling healthy eating behaviours; To teach healthy eating.

**Table 2 children-09-00510-t002:** Themes identified from parent and teacher interviews.

**Theme 1. Topics Discussed during Conversations**
1.a. Allergies
1.b. Insufficient food
1.c. Supervision requests
**Theme 2. Communication Methods and Barriers**
2.a. Communication channels
2.b. Communication barriers
2.c. Suggested ways to improve communication

**Table 3 children-09-00510-t003:** Parents’ communication from and expectations of their child’s classroom teacher.

**Part a. Communication between Parents and Teachers**	
Have you ever been contacted by your child’s classroom teacher regarding food or nutrition related issues? (Tick all that apply) (N = 784)	N (%)
I have not been contacted	559 (71)
When there is a food allergy issue in the classroom	161 (21)
Certain foods not allowed in my child’s lunch box	50 (6)
When my child does not eat his/her lunch	42 (5)
Other food related issues, please specify	42 (5)
Not enough food in my child’s lunch box	27 (3)
When would you contact your child’s classroom teacher regarding food and nutrition related issues? (Tick all that apply) (N = 772)	N (%)
When my child does not eat his/her lunch and needs supervision	278 (36)
I would never contact my child’s classroom teacher	246 (32)
When I do not want my child to be exposed to a particular food	230 (30)
Other food issues, please describe.	107 (14)
To ask what to put in the lunch box or whether a certain food is appropriate	101 (13)
to be sent to school	
**Part b. Parents’ expectations of teachers**	
To what extent do you agree or disagree with the following statements? (N = 787)	
I expect my child’s classroom teacher:	N (%) Agreement *
To teach healthy eating	698 (89)
To make sure my child is given enough time to eat lunch	696 (89)
To be a good role model by modelling healthy eating behaviours	620 (79)
To make sure school policies on healthy eating are always applied	452 (58)
To make sure my child eats all his/her lunch	201 (26)
To give feedback to me about the content of the lunch box	136 (17)
Which single action is the most important to you? (N = 787)	N (%)
To make sure my child is given enough time to eat lunch	330 (42)
To teach healthy eating	246 (31)
To be a good role model by modelling healthy eating behaviours	152 (19)
To make sure school policies on healthy eating are always applied	40 (5)
To make sure my child eats all his/her lunch	25 (3)
To give feedback to me about the content of the lunch box	4 (0)

* Note: The sum of ‘% Strongly agree’ and ‘% Agree’ is presented.

## Data Availability

The data presented in this study are available on request from the corresponding author. The data are not publicly available due to containing information that could compromise the privacy of research participants.

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
