# Peer review of "Parents’ Communication with Teachers about Food and Nutrition Issues of Primary School Students"

_children, 2022, doi:10.3390/children9040510_

Round 1

Reviewer 1 Report

The manuscript is extremely interesting and addresses a highly relevant subject the need to overcome communication barriers between parents and teachers and support teachers in their multifaceted professional roles, as well as improve the cooperation of parents and teachers has in the promotion of healthy eating habits among primary school children. This work ads data about the communication between these two groups regarding food and nutrition related issues.

The authors provided clear and relevant information about important issues to improve effectiveness in health promotion interventions, for instance. The manuscript is well written, informative and I have no concerns. Thus, it can be acceptable for publication.

Author Response

The authors thank Reviewer 1 for her/his supportive comments.

Reviewer 2 Report

Introduction

Lines 27-32: Only 5% of Australian children consume the daily recommended amount of fruit and vegetables, and around 40% of their daily energy comes from energy-dense, nutrient-poor food (Australian Bureau of Statistics, 2018b). In addition, one out of every four Australian children is overweight or obese (Australian Bureau of Statistics, 2018b), and unhealthy eating habits are known to be major contributors to these conditions (Australian Institute 31of Health and Welfare, 2011; Australian Institute of Health and Welfare, 2017). - Can you specify the age range of the children for this data? 

Line 40: primary school years- It might be helpful to include an age range in brackets for international readers that are not familiar with this terminology. 

Lines 46-48:  So, whilst parents are not the only role models for their children, they do control their children's food consumption by shopping, cooking, and providing food in the home environment (Reid et al., 2015).- Australian parents also provide food for their child to take to school so I think this needs to be acknowledged as well. 

In the field of health promotion, the stakeholder theory has recently gained prominence (Chen and Turner, 2012; Kok et al., 2015)- In this sentence or the next sentence an explanation is needed to explain how this content is relevant/connected to the previous paragraphs as it feel disconnected. 

The first two sentences (lines 60-65) could be deleted as this information is not that informative and repetitive. 

Lines 65-66: Frequently been recommended- By who? 

However, little is known about the current interactions and communications between parents and teachers regarding food and nutrition issues. - This statement is not particularly convincing since four studies are cited in the next sentence. 

In addition, a few studies have explored the interactions and communications of parents with teachers in preschool settings 
(Sharma et al., 2015; Finnane, 2017; Dinkel et al., 2021; Mena et al., 2020).- Not highly relevant for the study presented. 

However, parents’ and teachers’ communications and interrelationships regarding food and nutrition related issues is underexamined in primary school settings.- This statement is not consistent with the first sentence of this paragraph ( lines 73-75) as it was indicated that there is some evidence. Could you expand on this first sentence and explain the limitations of these studies and what evidence gap your study will be addressing? 

Methods:

Lines 98-99: ‘How would you describe your relationship and communication with 98 parents/teachers regarding food and nutrition related issues?- A closing quotation mark is missing. 

Lines 101-102: Parents and teachers were purposefully (Patton, 2002) recruited from Melbourne suburbs and regional areas of Victoria. - How many suburbs or what was the km radius included? 

Was the teacher's level of experience considered in the sampling? 

Lines 115-117: Because of the initial over-representation of government schools, eighteen parents and nine teachers were not selected and were informed by email. -How was this sample selected for exclusion?

Lines 120-121: The recruitment was finalised when the ‘conceptual depth’ (data saturation) point was reached (Nelson, 2013).- How was this determined?

How was the interview script/questions developed for study 1? 

Was there an other option for the second survey question for study 2? 

Results:

Line 207: Despite their different ethnic backgrounds- This needs more explanation. 

Was the sample gender balanced? 

Line 212: with a variety of teaching experience,- Can you describe the range of experience? 

Line 352: more than half were from Victoria (56%).- Can you explain why the sample ended up predominately coming from one state? 

How were the other responses interpreted in the survey results? 

Discussion 

-I would recommend deleting the first paragraph as it is not closely related to the study findings and is a distraction 

-Moreover, supporting classroom teachers  with a dedicated school health and wellness team would decrease teachers’ workload and increase communication effectiveness with parents.- I am not convinced that this recommendation is feasible. Has this strategy been used in other schools? 

-Another limitation that could be acknowledged is that the sample in the survey was predominately female. 

Author Response

The authors would like to express their thanks to the the reviewer 2 for her/his very helpful comments, which have helped to improve the quality of the paper.

Please see our responses to each of Reviewer 2’s suggestions below.

  1. Lines 27-32: Only 5% of Australian children consume the daily recommended amount of fruit and vegetables, and around 40% of their daily energy comes from energy-dense, nutrient-poor food (Australian Bureau of Statistics, 2018b). In addition, one out of every four Australian children is overweight or obese (Australian Bureau of Statistics, 2018b), and unhealthy eating habits are known to be major contributors to these conditions (Australian Institute 31of Health and Welfare, 2011; Australian Institute of Health and Welfare, 2017). - Can you specify the age range of the children for this data? 

Response 1:  The sentences  have been revised as ‘Only 5% of Australian children aged 2-17 years consume the daily recommended amount of fruit and vegetables, and around 40% of their daily energy comes from energy-dense, nutrient-poor food’ and ‘In addition, one out of every four Australian children aged 5-17 years is overweight or obese.. See lines 27 and 30. 

  1. Line 40: primary school years- It might be helpful to include an age range in brackets for international readers that are not familiar with this terminology. 

Response 2: It has been clarified in the manuscript as ‘Social factors include relationships with others, and in primary school years [age 5 to 12 years olds], parents and teachers are particularly influential.’  See line 40.

  1. Lines 46-48:  So, whilst parents are not the only role models for their children, they do control their children's food consumption by shopping, cooking, and providing food in the home environment (Reid et al., 2015).- Australian parents also provide food for their child to take to school so I think this needs to be acknowledged as well. 

Response 3: In addition, ‘ Australian parents also provide food for their children to take to school (Lawlis, 2016)’ has been added to the manuscript in lines 48 and 49.

  1. In the field of health promotion, stakeholder theory has recently gained prominence (Chen and Turner, 2012; Kok et al., 2015)- In this sentence or the next sentence an explanation is needed to explain how this content is relevant/connected to the previous paragraphs as it feel disconnected. 

Response 4: This paragraph has been moved to the Methods section. See lines 90-96.

  1. The first two sentences (lines 60-65) could be deleted as this information is not that informative and repetitive.

   Response 5: These two sentences have been deleted. See lines 62-67.

  1. Lines 65-66: Frequently been recommended- By who? 

Response 6: This sentence has been revised as ‘The cooperation of parents and teachers has frequently been recommended in the promotion of healthy eating habits among primary school children by researchers (Lindsay et al., 2006; Blom-Hoffman et al., 2008).  See line 69.

  1. However, little is known about the current interactions and communications between parents and teachers regarding food and nutrition issues. - This statement is not particularly convincing since four studies are cited in the next sentence. 

However, parents’ and teachers’ communications and interrelationships regarding food and nutrition related issues is underexamined in primary school settings.- This statement is not consistent with the first sentence of this paragraph ( lines 73-75) as it was indicated that there is some evidence. Could you expand on this first sentence and explain the limitations of these studies and what evidence gap your study will be addressing? 

 Response 7: The four studies cited are about the ways parents and teachers communicate with children regarding food and nutrition related issues. In contrast, the present study examines the way primary school parents and teachers communicate with each other.  

  1. In addition, a few studies have explored the interactions and communications of parents with teachers in preschool settings (Sharma et al., 2015; Finnane, 2017; Dinkel et al., 2021; Mena et al., 2020).- Not highly relevant for the study presented. 

Response 8: To the best of our knowledge, there have been no studies focussing on parent-teacher communications about food and nutrition related issues in primary schools, though as noted above we have mentioned studies in the preschool context.

We have revised the text as ‘In addition, a few studies in preschools have explored the interactions and communications of parents with teachers about food and nutrition issues (Sharma et al., 2015; Finnane, 2017; Dinkel et al., 2021; Mena et al., 2020). However, parents’ and teachers’ communications and interrelationships regarding food and nutrition related issues in primary schools have generally been underexamined. This paper aims to fill this gap.’ See lines 78-83.

To clarify further, we have also revised the sentence in the above paragraph as follows: ‘However, little is known about the current interactions and communications between parents and teachers regarding food and nutrition issues in primary schools’ See line 75.

Methods:

  1. Lines 98-99: ‘How would you describe your relationship and communication with parents/teachers regarding food and nutrition related issues?- A closing quotation mark is missing. 

Response 9: A closing quotation mark has been added in line 109.

  1. Lines 101-102: Parents and teachers were purposefully (Patton, 2002) recruited from Melbourne suburbs and regional areas of Victoria. - How many suburbs or what was the km radius included? 

Response 10: We didn’t decide on a km radius to be included. However, we selected parents from different suburbs of varying socioeconomic levels.

It has been revised as ‘Parents and teachers were purposefully (Patton, 2002) recruited from Melbourne suburbs of varying socioeconomic levels and regional areas of Victoria.  See line 112.

  1. Was the teacher's level of experience considered in the sampling? 

Response 11: No, it was not considered specifically but we did recruit teachers with various levels of experience. We reported their level of experience in another publication to demonstrate the characteristics of our sample  (Aydin et al. 2021). This reference has  now been  added to this manuscript in lines 235-236.

We have also included the information regarding teachers’ level of experience to the manuscript. See lines 228-231.

  1. Lines 115-117: Because of the initial over-representation of government schools, eighteen parents and nine teachers were not selected and were informed by email. -How was this sample selected for exclusion?

Response 12: More parents and teachers from government schools volunteered to take part in the interviews than was required. Therefore, they were not invited to participate and were informed by email. This has been added to the manuscript in lines 125-128.

  1. Lines 120-121: The recruitment was finalised when the ‘conceptual depth’ (data saturation) point was reached (Nelson, 2013).- How was this determined?

Response 13: This paragraph has been revised as ‘The recruitment was finalised when the ‘conceptual depth’ (data saturation) point was reached (Nelson, 2013). This is the point when further data gathering and analysis added little new to the conceptualisation (Corbin & Strauss 2008).  See lines 134-136.

  1. How was the interview script/questions developed for study 1? 

Response 14: The interview questions were original and based on a review of the literature in which gaps were identified. This information has been added to the manuscript. See lines 106-107 

  1. Was there an other option for the second survey question for study 2? 

Response 15: Yes, there was. Thanks for pointing this out. We have added this to the relevant table in line 198.

Results:

  1. Line 207: Despite their different ethnic backgrounds- This needs more explanation. 

Response 16: More explanation has been provided as below.

…..Despite their different ethnic backgrounds [13 parents were from Chilean, Turkish, Israeli, Japanese, Indonesian, New Zealander, Singaporean, Italian, Vietnamese, Polish, Persian, Swedish, Taiwanese backgrounds whereas six had an Australian background] (Lines 224-226).

  1. Was the sample gender balanced? 

Response 17: Sample was not gender balanced. This has been mentioned as a limitation in line 568.

  1. Line 212: with a variety of teaching experience,- Can you describe the range of experience? 

Response 18: The duration of teachers’ experience varied considerably; six had between zero and four years, three between five and nine years, three between ten and fourteen years, two between fifteen and nineteen years, and three had over nineteen years of experience.This has been added to the manuscript in lines 231-234.

  1. Line 352: more than half were from Victoria (56%).- Can you explain why the sample ended up predominately coming from one state? 

Response 19: The researchers who conducted the present study were based in Victoria; therefore, this might have facilitated recruitment of a higher number of parents from Victoria due to familiarity, personal networks and access to Deakin University’s social media profile (which is located in Victoria). This explanation has now been added in the manuscript in lines 552-555.

  1. How were the other responses interpreted in the survey results?

Response 20: In designing the question responses, we considered whether we had missed any significant answers that many parents would choose, hence we included the ‘other’ option. In response to the question ‘Have you ever been contacted by your child’s classroom teacher regarding food or nutrition related issues?’ parents who chose the ‘other’ option listed several reasons including praise for the prepared lunch, too much food in the lunchbox and sometimes teachers wanted to know about food prohibitions that may have existed at home. These points have been included in the text in lines 384-386

Most  parents who opted for the ‘other’ option for the question ‘When would you contact your child’s classroom teacher regarding food and nutrition related issues?’ mentioned food allergies. Other issues raised included lunchtime is being too short  and use of candies as rewards.  This has been clarified in lines 378-381.

Discussion 

  1. -I would recommend deleting the first paragraph as it is not closely related to the study findings and is a distraction 

Response 21: These two sentences have been deleted. See lines 435-441.

  1. -Moreover, supporting classroom teachers with a dedicated school health and wellness team would decrease teachers’ workload and increase communication effectiveness with parents.- I am not convinced that this recommendation is feasible. Has this strategy been used in other schools? 

Response 22: This month Victorian government has announced Healthy Kids Advisors scheme for local communities so this idea is likely to be  feasible in Victoria. ( https://www.kitchengardenfoundation.org.au/healthy-kids-advisors)

23.-Another limitation that could be acknowledged is that the sample in the survey was predominately female. 

Response 23: This limitation has been added as ‘Another limitation is that the sample was predominately female.’  See line 568.

References:

Aydin, G, Margerison, C, Worsley, A & Booth, A 2021, 'Parents’ and teachers’ views of the promotion of healthy eating in Australian primary schools', BMC Public Health, vol. 21, no. 1, pp. 1-12.

Corbin, J & Strauss, A 2008, Basics of Qualitative Research: Techniques and Procedures for Developing Grounded Theory vol. 12, Thousand Oaks, CA: Sage.